# Antimicrobial Activity and Chemical Constitution of the Crude, Phenolic-Rich Extracts of *Hibiscus sabdariffa, Brassica oleracea* and *Beta vulgaris*

**DOI:** 10.3390/molecules24234280

**Published:** 2019-11-24

**Authors:** Seham Abdel-Shafi, Abdul-Raouf Al-Mohammadi, Mahmoud Sitohy, Basma Mosa, Ahmed Ismaiel, Gamal Enan, Ali Osman

**Affiliations:** 1Botany and Microbiology Department, Faculty of Science, Zagazig University, Zagazig 44519, Egypt; shefaalab1984@gmail.com (B.M.); ahmedismaiel80@gmail.com (A.I.); gamalenan@ymail.com (G.E.); 2Department of Science, King Khalid Military Academy, Riyadh 11495, P.O. Box 22140, Saudi Arabia; 3Biochemistry Department, Faculty of Agriculture, Zagazig University, Zagazig 44511, Egypt; mzsitohy@hotmail.com (M.S.); ali_khalil2006@yahoo.com (A.O.)

**Keywords:** *Hibiscus sabdariffa*, *Brassica oleracea*, Beta vulgaris, crude phenolic rich extract, anthocyanins, GC–MS analysis, antimicrobial

## Abstract

Crude, phenolic-rich extracts (CPREs) were isolated from different sources, such as *Hibiscus sabdariffa (H. sabdariffa), Brassica oleracea* var. capitata f. rubra *(B. oleracea)* and *Beta vulgaris* (*B. vulgaris)* and characterized. These CPREs showed potential antibacterial and antifungal activities. *H. sabdariffa* CPRE (HCPRE) is the most potent, as it inhibited all tested bacteria and fungi. Total anthocyanins content (TAC), total phenolic content (TPC) and total flavonoid content (TFC) were estimated in all three CPREs. *H. sabdariffa* contained 4.2 mg/100 g TAC, 2000 mg/100 g of TPC and 430 mg/100 g of TFC in a dry weight sample. GC–MS analysis of HCPRE showed 10 different active compounds that have antimicrobial effects against pathogenic bacteria and fungi, especially alcoholic compounds, triazine derivatives and esters. Scanning and transmission electron microscopy images of *Staphylococcus*
*aureus* DSM 1104 and *Klebsiella pneumonia* ATCC 43816 treated with HCPRE (50 μg/mL) exhibited signs of asymmetric, wrinkled exterior surfaces, cell deformations and loss of cell shapes; and adherence of lysed cell content led to cell clumping, malformations, blisters, cell depressions and diminished cell numbers. This indicates death of bacterial cells and loss of cell contents. *Aspergillus ochraceus* EMCC516 (*A. ochraceus*, when treated with 100 μg/mL of HCPRE showed irregular cell organelles and cell vacuolation.

## 1. Introduction

The demand for effective natural antimicrobial compounds free of toxicity and environmental hazards has enormously increased as a result of the mounting increased drug resistant bacteria, nullifying drugs’ effectiveness and causing widespread infections [1]. To avoid the increasingly growing antibiotic resistance, many natural products such as native or modified proteins have been investigated for their antibacterial actions as possible substitutes for the antibiotics [2,3,4,5,6,7,8,9,10,11,12,13]. Pathogenic bacteria and fungi affect agriculture, food industry, consumers and the national economy. The safe, plant-derived compounds with antimicrobial activity against pathogens are vital. For instance, carvacrol and cinnamaldehyde reduced *Campylobacter jejuni* and *Salmonella enterica* to undetectable levels at 0.2% concentration. The native cowpea seed proteins 7S and 11S were reported to strongly inhibit the in vitro growth of *Pseudomonas aeruginosa* ATCC 26853 and *Salmonella typhimurium* ATCC 14028 [14]. Additionally, soybean’s glycinin basic subunit was able to inhibit methicillin-vancomycin intermediate *Staphylococcus aureus* (MRSA-VISA) while soy glycinin was competent to impede *Bacillus* spore germination [15,16].

Anthocyanins are the most important group of water-soluble pigments in nature. The word “Anthocyanin” is derived from two Greek words ‘anthos’ meaning flower and ‘kyanos’ meaning dark blue, referring to its important role as a natural colorant [17,18]. Anthocyanins are the polyphenolics that are responsible for red to purple color in plants. They are members of flavonoid group of phytochemicals [19,20]. Primary constituents that are present in flavonoid group are anthocyanins, flavanols, flavones, flavanones, etc. Anthocyanins are the hydroxyl and methoxyl derivatives of phenyl-2-benzopyrylium salts, regarded as flavonoid compounds [21].

The previous study reported two major anthocyanins, delphinidine-3-sambubioside and cyanidine-3-sambubioside, and two minor compounds, i.e., delphinidine-3-glu-coside and cyanidine-3-glucoside, present in the calyces of *Hibiscus sabdariffa* (roselle) [22]. Approximately 85% of anthocyanins were delphinidine-3-sambubioside which is the principal source of the antioxidant capacity of roselle extract [23].

The phenolic structure of anthocyanin stands behind their antioxidant activity; i.e., their capability to scavenge reactive oxygen species (ROS); i.e., superoxide (O_2_^−^), singlet oxygen (O_2_), peroxide (ROO), hydrogen peroxide (H_2_O_2_), and hydroxyl radicals (OH) [24]. The herbs’ antioxidant activities may be attributed to the plant pigments constituting the major components of the herbal extract. Antioxidant assays in foods and biological systems can be classified in two groups, those based on the evaluation of lipid peroxidation, and those based on the measurement of free radical scavenging power [25,26].

Roselle is widely used for treating diseases. The aqueous methanolic extract of roselle was analyzed for its phytochemical constituents, antimicrobial activity and cytotoxicity, revealing the following components, cardiac glycosides, flavonoids, saponins and alkaloids. It exhibited in vitro antibacterial activities against *Staphylococcus aureus*, *Bacillus stearothermophilus*, *Micrococcus luteus*, *Serratia mascences*, *Clostridium sporogenes*, *Escherichia coli*, *Klebsiella pneumoniae*, *Bacillus cereus* and *Pseudomonas fluorescence* [27].

The in vitro antimicrobial action of roselle extract was ascribed to the flavonoids, which can establish complexes with the bacterial cell walls, enhancing their permeation to the extract. The mechanism of action may include some metabolic steps, e.g., inhibition of electron transport protein translocation, phosphorylation steps, and some other enzyme dependent reactions ending with raised membrane permeability coupled with the leakage of the bacterial cell constituents [28].

Red cabbage (*Brassica oleracea* L.) has been extensively studied, due to its distinct color and potential physiological functions, arising probably from the presence of anthocyanin [29], the major pigment of this plant [30], which is composed of cyanidin-3-diglucoside-5-glucoside “cores,” that are non-acetylated, mono-acetylated or di-acetylated with p-coumaric, caffeic, ferulic and sinapic acids. Anthocyanin was previously extracted from red cabbage using high pressure CO_2_ [31]. Red cabbage is one of the most important vegetables belonging to the family *Cruciferae*. It is an herbaceous plant characterized by a short stem crowned up with a mass of red leaves (head). It is mainly used as salad, but can be cooked or pickled. Red cabbage is known for its medicinal properties; e.g., anticancer activity, due to the presence of indole-3-carbinol. It is an excellent source of vitamin C, vitamin B complex, potassium and calcium. The purple/red color leaves are due to a pigment belonging to anthocyanins (flavins). This color varies according to the soil pH, being more reddish in acidic soils, purple in neutral soils and greenish yellow in alkaline soil. Red cabbage is a rich source of natural antioxidants such as ascorbic acid, α-tocopherol, β-carotene and lutein [32]; oligosaccharides; and a some bioactive substances; e.g., flavonols and glucosinolates [33]. Its wide spread use in traditional medicine were ascribed to its antioxidant, anti-inflammatory and antibacterial properties. It is used for treating symptoms associated with gastrointestinal disorders; e.g., peptic and duodenal ulcers, gastritis or irritable bowel syndrome [34].

Natural colorants may be promising active biological agents. For example, phycocyanins were found to have many biological activities [35,36,37]. Likewise, red beet (*Beta vulgaris* L.) grows red or purple tuberous root vegetables, known as beetroot or garden beets, which are a firm, clean, globe-shaped vegetable with no mucilaginous or watery tissues, and its tubers contain freshly emerged young leaves. The biological importance of red beet is based on its high red pigment content, (betalain), which displays excellent values, meeting some applications in food and pharmaceutical products. Among many plants accumulating betalains, only red beet and prickly pear (*Opuntia ficus-indica*) are approved for food and pharmaceutical applications [38]. For example, the use of beet extract as a food colorant is approved by the US Food and Drug Administration (FDA). As a powerful antioxidant pigment, betanin may provide protection and reduce risk of cardiovascular disease and cancer [39].

Based on the potentially high content of anthocyanins and other bioactive compounds, three plants growing in Egypt were selected for this study; *Hibiscus sabdariffa (H. sabdariffa), Brassica oleracea (B. oleracea)* and *Beta vulgaris* (*B. vulgaris),* as the sources for isolating the crude phenolic rich extract (CPRE). These extracts (CPRE) were analyzed for total phenolics content and total flavonoids and evaluated for their antibacterial and antifungal activity by different methods.

## 2. Results

### 2.1. Chemical Characterization of Isolated CPRE

#### 2.1.1. Total Anthocyanin, Total Phenolic and Total Flavonoid Contents

Total phenolic contents (TPC) of all samples were assayed by Folin–Ciocalteu’s method, and found to be varied (Table 1). The highest amount of TPC was observed in *H. sabdariffa* CPRE (HCPRE) (2000 mg GAE/100 g dry pigment). *B. oleracea* showed the lowest amount of TPC (150 mg GAE/100 g dry pigment). The highest amounts of anthocyanin and flavonoid contents were observed with HCPRE.

#### 2.1.2. Gas Chromatography-Mass Spectrometry (GC–MS) Analysis of HCPRE

The chemical compounds extracted from HCPRE (2000 µg/mL) were obtained by GC–MS analysis (Table 2; Figure 1). HCPRE contains ten active compounds, most of which have antimicrobial effects against pathogenic bacteria and fungi, especially alcoholic compounds, triazine derivatives, mercepto compounds and esters.

### 2.2. Antimicrobial Activity of Crude Phenolic Rich Extract (CPRE) (2000 µg/mL) against Pathogenic Bacteria

The CPREs from H. sabdariffa, Brassica oleracea var. capitata f. rubra and B. vulgaris (2000 μg/mL) were tested for their antibacterial actions against S. aureus, Streptococcus pyogenes, Listeria monocytogenes, E. coli, K. pneumonia and Pseudomonas aeruginosa (Table 3). HCPRE exhibited the highest inhibition zones against the all bacteria, but B. vulgaris pigment had lower inhibition zones.

### 2.3. Minimum Inhibitory Concentrations (MICs) Values of HCPRE and B. oleracea Pigments against Bacteria

Different concentrations of extracted HCPRE were prepared (0, 25, 50, 100, 200 and 250 μg/mL) and tested for their antibacterial action (Table 4, Appendix A). The results indicated that the MIC of the pigment against Gram positive *S. aureus, S. pyogenes* and *L. monocytogenes* was 50 μg/mL; the MIC against Gram negative *E. coli* and *P. aeruginosa* was 25 μg/mL; and it was 50 μg/mL for *K*. *pneumoniae*. *S*. *aureus* and *K*. *penuomonia* are the most sensitive bacteria to HCPRE. Different concentrations of extracted *B. oleracea* pigment (BOP) were prepared (0, 25, 50, 100, 200 and 250 μg/mL) and tested for their antibacterial action against pathogenic bacteria (Table 4). The results showed that the MICs of the BOP against *S. aureus, S. pyogenes* and *L. monocytogens* were 25,100 and 50 μg/mL; and against *E. coli, K*. *pneumonia* and *P. aeruginosa* were 100, 200 and 100 μg/mL respectively. *S*. *aureus* and *P. aeruginosa* are the most sensitive bacteria to BOP.

### 2.4. Antifungal Activity of HCPRE against Pathogenic Fungi and MIC Values

HCPRE (2000 μg/mL) strongly inhibited all tested fungi (Table 5). Different concentrations of HCPRE extract were prepared (0, 100, 200, 300, 400 and 500 μg/mL) and tested for their antifungal actions against pathogenic fungi. The results showed that the MICs of the pigment against the fungi (*A*. *ochraceus*, *F. oxysporum*, *P. expansum* and *P. citrinum*) were 100 μg/mL, and according to the diameter of inhibition zones it showed that *A. ochraceus* was the most sensitive fungus to HCPRE (Table 6; Appendix A). Moreover, different concentrations of extracted *B. oleracea* pigment were prepared (0, 100, 200, 300, 400 and 500 μg/mL) and tested for their antifungal actions against pathogenic fungi.

The results showed that the MICs of BOP against *A*. *ochraceus* and *F. oxysporum* were 400 and 300 μg/mL, respectively. BOP do not inhibit the growth of *P. expansum* and *P. citrinum* (Table 6, Appendix A).

### 2.5. Quantitative Inhibition of Pathogenic Bacteria by Plant Pigments (Bacterial Growth Curve)

Plant pigments (*H. sabdariffa* and *B. oleraceae*) were added at their MIC values to test tubes containing 10 mL NB and inoculated with 10 µl aliquots of bacterial suspensions. Samples and untreated test tubes (controls) were incubated at 37 °C for 30 h. At appropriate time intervals, 1 mL aliquots of bacterial suspensions were withdrawn and were analyzed for their turbidity at OD600. Results are given in (Figure 2).

In the case of treating with *H. sabdariffa* pigment, almost no growth was shown in bacterial test tubes treated with pigment. However, bacteria grew rapidly in control tubes (without pigment) and turbidity went from 0.1 to almost 1.2 at OD600. Distinctive inhibition was observed at OD600, which increased only ≥0.0 in all of them. Moderate inhibition was observed for growth recorded at OD600 within 30 h in contradicting situations. In case of BOP bacteria grew rapidly in control tubes and bacterial growth inhibited in treated tubes.

### 2.6. SEM and TEM Microscopy Analysis

SEM images showed that the presence of HCPRE (50 μg/mL) on NB media containing *S*. *aureus* affected the bacterial cells and caused cell deformations, wrinkles and loss of cell shapes. The adherence of lysed cell content led to cell clumping, and this was seen after 18 h of incubation at 37 °C. HCPRE (50 μg/mL) on nutrient broth (NB) media containing *K*. *pneumonia* showed malformations (increases in length and decrease in width), cell depressions, diminished cell number and observed rectangular cells as loss of regular cell shapes were detected. That indicates the death of cells and loss of cellular contents. SEM images of *A*. *ochraceus* treated with HCPRE (100 μg/mL) showed destruction of conidia, failing in conidia formation, thinning and condensation of mycelia; then, malformation and loss of cell contents.

TEM images showed that *S*. *aureus* and *K*. *pneumonia* affected by HCPRE (50 μg/mL) on NB media showed malformed shapes, cell depress, cell vacuolation, blisters and wrinkles.

*A*. *ochraceus*, when treated with (100 μg/mL) of HCPRE showed irregular cell organelles and cell vacuolation (Figure 3A–F).

## 3. Discussion

Natural colorants obtained from vegetables are more available and healthy than synthetic colors [40]. The natural pigments are used in medicine and food [41]. Many bacterial organisms have developed increasing resistance against the frequently used antibiotics [42].

In this study, the pigments extracted from *H. sabdariffa* inhibited all tested bacteria and fungi. The previous studies showed that *H. sabdariffa* inhibited *S*. *aureus*, *B. cereus, E. coli*, *Clostridium sp., Klebsiella pneumonia and Pseudomonas fluorescens* [42]. Herbal drug formulations composed of medicinal plants have been inherited from ancient times to treat many diseases, since their antimicrobial properties suggest them as potentially rich sources of various potent drugs [43]. Natural antimicrobials have enormous therapeutic potential, since they can probably conduct the required functions without any posing health hazards often associated with synthetic agents [44]. *H. sabdariffa*’s aqueous extract has strong activity against *C. albicans* [27]. Roselle can be utilized either as a distinct functional food or as an active ingredient in other functional food potentially applicable in the treatment of various degenerative diseases [45].

Based on the results, the antibacterial action of anthocyanin was concentration-dependent. HCPRE contains total anthocyanin content of 4.2 (mg/100 g) in dry pigment. Anthocyanin had relatively higher antibacterial activity than antifungal activity against the microorganisms investigated. Anthocyanins were reported to have anticarcinogenic activity against multiple cancer cell lines in vitro and in vivo tumor types [46]. *H. sabdariffa* showed antimicrobial activities against some food pathogenic microbial isolates, e.g., *E. coli* O157:H7, *Salmonella enterica* and *L. monocytogenes*, as well as veterinary, and clinical isolates. This indicated that HCPRE extract is broadly effective against different microorganisms, suggesting its application as a potential food-grade antimicrobial [28]. The antibacterial effects of roselle calyx aqueous and ethanol extracts and protocatechuic acid against food spoilage bacteria *Salmonella typhimurium* DT104, *E. coli* O157:H7, *L. monocytogenes*, *S. aureus* and *B. cereus* were examined by [47]. The inhibitory activities in a dose-dependent manner against bacteria in ground beef and apple juice were studied, and it was suggested that they might be potent agents as food additives to prevent contamination from these bacteria.

The anthocyanins and polyphenols from the *H. sabdariffa* (roselle) were extracted by an aqueous or organic solvent. The dried roselle contained total anthocyanins as cyanidine 3-glucoside 622.91 mg/100 g and 37.42 mg/100 g total phenolic content in dry weight samples [48]. A recent study identified delphinedine-3-*O*-sambubioside, delphinidine-3-*O*-glucoside and cyani-dine-3-*O*-sambubioside at the concentrations of 7.03 mg/g, 1.54 mg/g and 4.40 mg/g in the roselle extract. GC–MS analysis showed 10 compounds in HCPRE. All of them have previously been shown to have antimicrobial activity. It is quite known that many *Hibiscus* species contain different classes of secondary metabolites, including flavonoids, anthocyanins, terpenoids, steroids, polysaccharides, alkaloids, sesquiterpene, quinones and naphthalene groups. Some of these components have antibacterial, anti-inflammatory, antihypertensive, antifertility, hypoglycemic, antifungal and antioxidative activities [49]. The antioxidant capacity of anthocyanins is dependent on its basic structural orientation; i.e., the ring orientation will determine the readiness of a hydrogen atom from a hydroxyl group to be donated to a free radical and the capability of the anthocyanin to support an unpaired electron [25]. *H. sabdariffa* is a safe medicinal plant, having medical compounds with nutritional and medicinal properties [50].

In this study, *S*. *aureus* and *K*. *pneumonia* were affected by HCPRE (50 μg/mL), showing malformed shapes, cell depressions, cell vacuolation, blisters and wrinkles. *A*. *ochraceus*, when treated with (100 μg/mL) of HCPRE, showed irregular cell organelles. The anthocyanin-rich blueberry extract was capable of inhibiting the growth, adhesion and/or biofilm formation of all of the following: *P. aeruginosa*, *E. coli*, *P. mirabilis*, *A. baumannii* and *S. aureus* [51]. Roselle contains proanthocyanidins which combine or transform the structural entity of P-fimbriae of bacterial cells; thus, inhibiting their adhesion to the ur-epithelium and formation of biofilms in vitro [45]. The antimicrobial properties of eight food dyes against 10 bacteria and five fungal organisms were previously investigated, showing that the red dyes were associated with the best antibacterial activities, while the yellow ones were more linked to better antifungal activity. Besides the antimicrobial analysis, antioxidant activity, measured by three different methods, was also investigated. In all the methods, red dye was found to have greater antioxidant activity. It suggests that the addition of these dyes in food not only enhances the value addition by making the food more presentable but also shall address the issue of food supplementation with substances that are good antibiotics and antioxidants, subsequently proving to be health benefactors [52].

## 4. Materials and Methods

### 4.1. Crude Phenolic Rich Extract (CPRE) Preparation

All chemicals used in this work were supplied from Al-Gomhorya Company for chemicals and public procurement Zagazig city, Egypt. The calyx of *H. sabdariffa*; the leaves of *Brassica oleracea* var. capitata f. rubra *(B. oleracea* ) and *B. vulgaris* tuberous root vegetables were obtained from local market in Egypt). The plants we used were botanically classified by Samer Teleb, Taxonomy and Flora, Botany and Microbiology Department, Faculty of Science, Zagazig University, Egypt. Milled sample (0.5 g) was soaked in 50 mL of ethanol for 24 h and the extract obtained was pre-filtered with Whatman No.4 filter (Whatman® Prepleated Qualitative Filter Paper, Grade 4V, Sigma-Aldrich, USA) before evaporation using a vacuum rotary evaporator (BüCHI-water bath-B-480, Czech Republic) at 30 °C.

### 4.2. Crude Phenolic Rich Extract Characterization

#### 4.2.1. Determination of Anthocyanins

Total anthocyanin content was colorimetrically determined according to the procedure described by [53] where a known volume of the filtered extract was diluted to 100 mL with the extracting solvent and the resulting color was measured at 520 nm for water and citric acid solution extracts and at 535 nm for acidified ethanol using Spectrophotometer (JENWAY-6405 UV/VIS, Chelmsford, England). The total anthocyanin content defined as cyanidin-3-glucoside was calculated using the following Equation (1):(1)$$\mathrm{Total}\;\mathrm{anthocyanins}\;\left(\frac{\mathrm{mg}}{100\;g}\right)=\frac{\mathrm{Absorbance}\;\times \;\mathrm{dilution}\;\mathrm{factor}}{\mathrm{Sample}\;\mathrm{weight}\;\times \;55.9}\times 100.$$

#### 4.2.2. Determination of Total Phenolic Compounds (TPCs)

The TPCs were estimated by Foline–Ciocalteu reagent as described by [54]. One milliliter of sample (1000 µg in 1 mL) was added to 5 mL of Folin–Ciocalteu reagent (diluted with water 1:10, v/v) and 4 mL sodium carbonate (75 g/L). The tubes were vortex mixed for 15 s and left stand 30 min at 40 °C, before measuring the absorbance of the developed color at 765 nm. Gallic acid was used to establish the standard curve (20–200 µg/ mL). The extent of reducing of the Folin–Ciocalteu reagent by the sample was expressed as mg of gallic acid equivalents (GAE) per g of extract. The calibration equation for gallic acid was y = 0.001x + 0.0563 (R^2^ = 0.9792), where y and x are the absorbance and concentration of gallic acid in µg/mL, respectively.

#### 4.2.3. Total Flavonoids (TFs) Determination

Total flavonoids (TFs) were estimated according to the protocol of [55] by blending 2 mL aliquot of 20 g/L AlCl_3_ ethanol reagent with 1 mL of the extract (1000 µg in 1 mL solvent) and measuring the developed color absorbance at 420 nm after 60 min. Quercetin was used to establish the standard curve (20–200 µg/mL) and total flavonoid content was expressed as quercetin equivalent (QE), based on the standard curve. The calibration equation for quercetin was y = 0.0012x + 0.008 (R^2^ = 0.944), where y is absorbance and x is concentration of quercetin in µg/mL.

#### 4.2.4. Gas Chromatography–Mass Spectrometry (GC–MS) Analysis

The chemical composition analysis of the samples was carried out using Trace GC1310-ISQ mass spectrometer (Thermo Scientific, Austin, TX, USA) with a direct capillary column TGram negative5MS (30 mm× 0.25 mm × 0.25 µm film thickness, Thermo Scientific, Austin, TX, USA). The column oven temperature was initially held at 50 °C; then increased by 7 °C/min increments to 200 °C hold for 2 min; and the final temperature at 290 °C was reached by 15 °C/min increments and held for 2 min. The injector and MS transfer line temperatures were kept at 270 and 250 °C, respectively. Helium, the carrier gas, was pumped at a constant flow rate of 1 mL/min. The solvent delay was 3 min, and 1 µL aliquots of the diluted samples were injected automatically using an Autosampler AS1300 coupled (Thermo Scientific, Austin, TX, USA) with GC. The ion source temperature was set at 200 °C. EI mass spectra were collected at 70 eV ionization voltages over a range of *m*/*z* 45–400 within full scan mode. The chemical composition of the obtained components was concluded by comparing their retention times and mass spectra with those of WILEY 09 and NIST 11 mass spectral database.

### 4.3. Collection of Pathogenic Bacteria and Fungi

*S. aureus* DSM 1104, *St. pyogenes* ATCC 19615, *L. monocytogenes* LMG10470, *E. coli* LMG 8223, *K. pneumonia* ATCC 43816 and *P. aeruginosa* LMG 8029 were used. Also, pathogenic fungi such as *A. ochraceus* EMCC516 were obtained from Egyptian Microbial Culture Collection (Microbiological Resoures Center MIRCEN, Cairo, Egypt); other fungi were *F. oxysporum*, *p. citrinium* and *P. expansum*. All bacteria and fungi used in this study were kindly offered by Botany and Microbiology Department (Laboratory of Bacteriology and Laboratory of fungi), Faculty of Science, Zagazig University, Zagazig, Egypt. Stock bacterial cultures were routinely kept at −20 °C in glass beads and were sub-cultured and propagated in brain heart infusion broth (BHIB) (Oxoid). Slope cultures were prepared fresh on nutrient agar for every experiment [11] and stored at 4 °C throughout the experimental work.

### 4.4. Antibacterial and Antifungal Activities of the CPREs

The antibacterial and antifungal activities of CPREs (2000 μg/mL) were tested against the experimental pathogenic bacteria and fungi by agar well-diffusion assays [56].

### 4.5. MIC Values of H. Sabdariffa CPRE

Pure cultures of bacterial strains were sub-cultured on BHIB at 37 °C. Each strain was spread uniformly onto an individual plate with a sterile cotton swab. Uniform wells (6 mm diameters) were made on nutrient agar (NA) plates using a gel puncturing tool. Aliquots (50 μL) of pigment solutions (0, 25, 50, 100, 200 and 250 μg/mL) were placed into each well. Sterilized distilled water was considered the negative control. After 24 h incubation at 37 °C, the diameters of the inhibition zones (mm) were measured using a transparent millimeter ruler. The pure cultures of fungal strains were sub-cultured on yeast extract agar (YES) at 30 °C. Each strain was spread uniformly onto the individual plates using sterile cotton swabs. Wells of 6 mm diameter were similarly made on YES plates. Aliquots (50 μL) of HCPRE and *B. oleracea* pigment solutions (0, 100, 200, 300, 400 and 500 μg/mL) were placed in each well. After 4 days’ incubation at 30 °C, the diameters of inhibition zones (mm) were similarly measured.

### 4.6. Quantitative Inhibition of Pathogenic Bacteria by CPRE (Bacterial Growth Curve)

A series of test tubes each containing 10 mL of nutrient broth (NB) were inoculated with 100 µL of log phase bacterial suspension and were then treated with 50 μg/mL HCPRE for all bacteria; and 200 μg/mL *B*. oleracea pigment for *E*. *coli* and 100 μg/mL for other bacteria. Control test tubes contained NB with bacteria only. Samples and controls were incubated at 37 °C. Growth was determined at time 0 and after 6, 12, 18, 24, 30, 36, 42 and 48 h of incubation by the turbidity method (OD600) using a spectrophotometer (JENWAY-6405 UV/VIS, Chelmsford, England).

### 4.7. Scanning and Transmission Electron Microscopy (SEM-TEM)

*S*. *aureus* (Gram positive bacteria) and *K*. *pneumonia* (Gram negative bacteria) were selected for scanning electron microscopy (SEM) and transmission electron microscopy (TEM). Bacteria were grown on NB media and incubated at 37 °C to reach maximum level of 10^6^ CFU/mL. The MIC values of about 50 μg/mL of HCPRE were added to *S. aureus* and *K*. *pneumonia* plates except for controls and incubated at 37 °C for 18 h. Also *A. ochraceus* was grown on YSA and incubated at 30 °C for 3 days to reach the maximum level of growth and MIC value of about 100 μg/mL of *H*. *sabdariffa* pigment.

#### 4.7.1. Scanning Electron Microscopy (SEM)

SEM (JEOL-scanning electron microscope JSM-6510 L.V SEM-JAPAN) at electron microscope (EM) Unit, Mansoura University, Egypt was used to evaluate the morphological changes of tested microorganisms as described in [2,16].

#### 4.7.2. Transmission Electron Microscopy (TEM)

TEM (JEOL JEM -2100, JAPAN) at EM Unit, Mansoura University, Egypt was used to evaluate ultrastructural changes of tested microorganisms as described in [14,57].

### 4.8. Statistical Analysis

The collected data were tabulated and analyzed using IBM SPSS software (version 26, IBM corporation, Chicago, IL, USA). The results were expressed as a means ± standard errors (SEs) in either tables or figures.

## 5. Conclusions

According to the obtained results, it can be concluded that *H. sabdariffa* pigment could be used as an antibacterial and antifungal agent. It can be efficiently and successfully used as safe, natural products. It can be prepared with low costs.

## Figures and Tables

**Figure 1 molecules-24-04280-f001:**
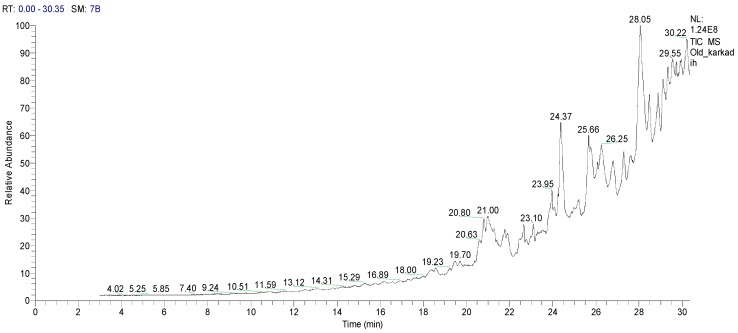
The TIC chromatogram of *H*. *sabdariffa* using GC–MS. RT—retention Time; SM—signal in method; NL—noise level.

**Figure 2 molecules-24-04280-f002:**
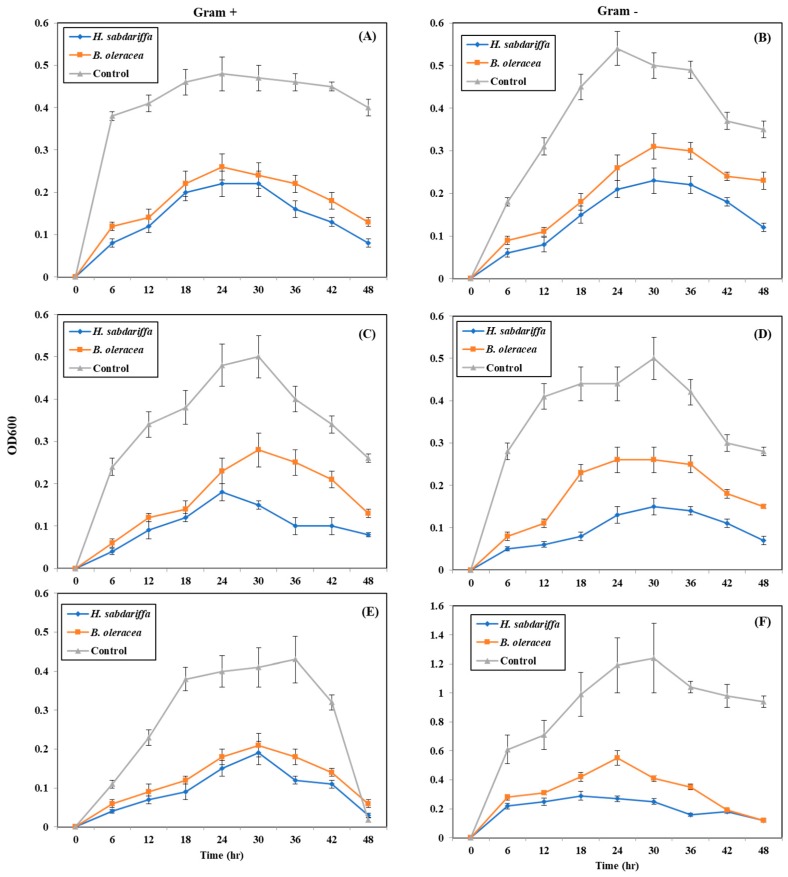
Quantitative inhibition of Gram-positive and Gram-negative bacteria by MIC of *H*. *sabdariffa* crude phenolic rich extract. (**A**) *S. aureus*; (**B**) *E. coli*; (**C**) *St. pyogenes*; (**D**) *K. pneumonia*; (**E**) *L. monocytogenes*; (**F**) *P. aeruginosa*.

**Figure 3 molecules-24-04280-f003:**
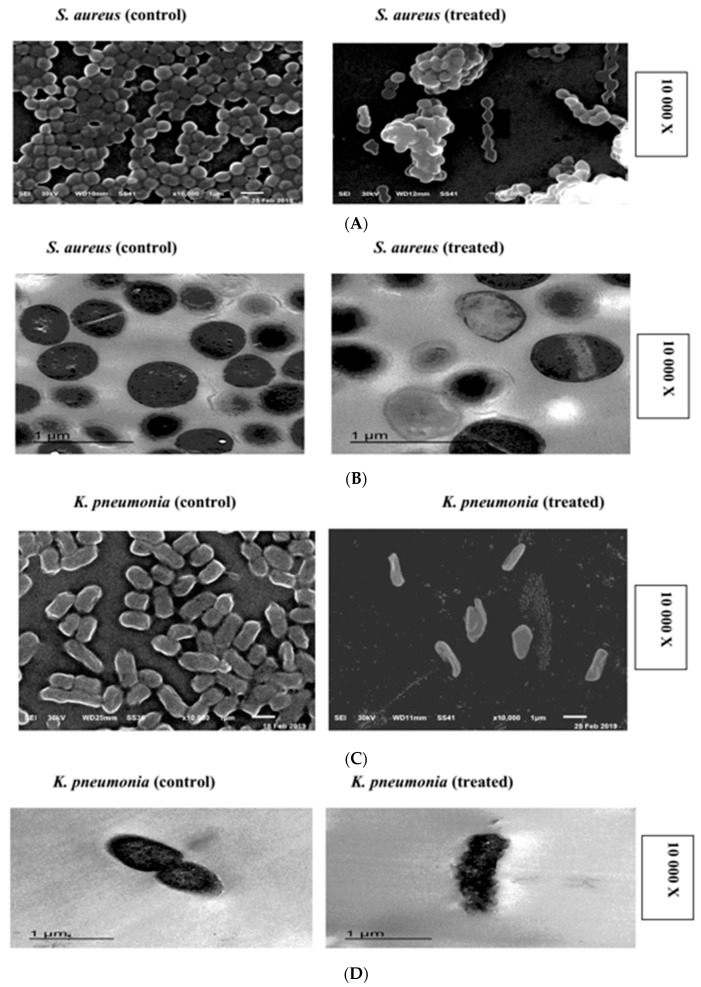
(**A**) SEM of *Staphylococcus aureus* affected by 50 µg/mL of HCPRE. (**B**) TEM of *S. aureus* affected by 50 µg/mL of HCPRE. (**C**) SEM of *K. pneumonia* affected by 50 µg/mL of HCPRE. (**D**) TEM of *K. pneumonia* affected by 50 µg/mL of HCPRE. (**E**) SEM of *A. ochraceus* affected by 100 µg/mL of HCPRE. (**F**) TEM of *A. ochraceus* affected by 100 µg/mL of HCPRE.

**Table 1 molecules-24-04280-t001:** Chemical characterization of isolated pigments.

Samples	Total Anthocyanin Content (mg/100 g Dry Pigment)	Total Phenolic Content (mg GAE/100 g Dry Pigment)	Total Flavonoid Content (mg QE/100 g Dry Pigment)
*H. sabdariffa*	4.2	2000	430
*B. oleracea*	2.7	150	50
*B. vulgaris*	3.8	400	120

GAE: gallic acid equivalent. QE: quercetin equivalent.

**Table 2 molecules-24-04280-t002:** The chemical compounds in the *Hibiscus sabdariffa* pigment (crude, phenolic-rich extracts—HCPRE) extracted, analyzed by GC–MS.

No	Classification	M. Formula	M. W.	Compound Name and Structure
1	Hydrocarbons (Alkan)-saturated compounds	C_22_H_66_	450	CH_3_(CH_2_)_3_OCH_3_—Dotriacontan
2	Alcoholic componds	C_17_H_36_O	256	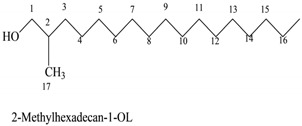
3	Triazine derivatives	C_5_H_8_ClN_5_	173	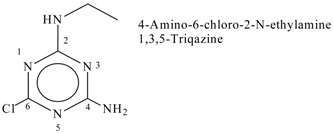
4	Unsat. alcoholic compound	C_19_H_38_O	280	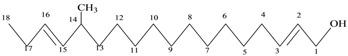 14-Methyl-2,15-octadecadien-1-OL
5	Unsaturated ester	C_17_H_22_O_2_	268	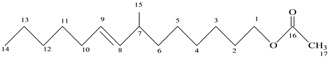 7-Methyl-2-tetradecan-1-OL acetate
6	Mercepto compound	C_16_H_34_S	258	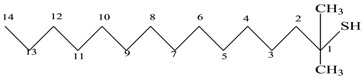 1,1-Dimethyl tetradecyl hydrosulfide or tert-headecanethiol (com.)
7	Alkenes	C_19_H_38_	266	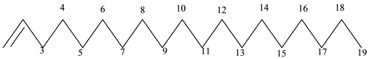 1-Nonadecene or Monadeca-1-ene
8	Primary alcohols	C_37_H_76_O	536	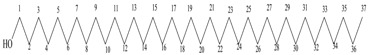 1-Heptalriaontanol or Heptatricotanol
9	Unsaturated ester	C_17_H_30_O_2_	266	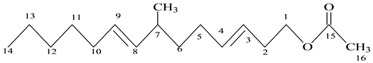 7-Methyl hexadeca-3,8-dienoate
10	Natural product (Cholesterol)	C_28_H_48_O	400	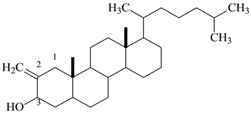 2-Methylene cholestan-3-oL

**Table 3 molecules-24-04280-t003:** Antibacterial activity of crude phenolic rich extracts (CPREs) (2000 µg/mL) from three plants against pathogenic bacteria using agar well diffusion assays.

Microorganisms	Inhibition Zone (mm)		
	*H. sabdariffa*	*B. oleracea*	*B. vulgaris*
Gram positive bacteria			
*S. aureus*	48 ± 8.0	38 ± 3.0 ^a^	24 ± 4.0 ^a^
*St. pyogenes*	40 ± 5.0 ^b^	32 ± 2.0 ^b^	10 ± 2.0 ^b^
*L. monocytogenes*	32 ± 4.0 ^c^	37 ± 3.0 ^a^	13 ± 3.0 ^b^
Gram negative bacteria			
*E. coli*	46 ± 6.0 ^a^	28 ± 0.50 ^c^	7 ± 0.50 ^c^
*K. pneumonia*	48 ± 6.5 ^a^	30 ± 2.0 ^b^	18 ± 2.0 ^a^
*P. aeruginosa*	32 ± 3.0 ^c^	29 ± 0.50 ^c^	7 ± 0.50 ^c^

Means in the same column having different letters are significantly different (*p* ≤ 0.05).

**Table 4 molecules-24-04280-t004:** Minimum Inhibitory Concentrations (MIC) values of HCPRE and *Brassica oleracea* pigments against pathogenic bacteria using agar well diffusion assays.

Microorganisms	Inhibition Zone (mm)					
	0	25	50	100	200	250
*H. sabdariffa* pigment						
Gram positive bacteria						
*S. aureus*	−ve	−ve	15 ± 3.0 ^c^	20 ± 2.0 ^b^	23 ± 3.0 ^b^	28 ± 4.0 ^a^
*St. pyogenes*	-ve	−ve	9 ± 0.5 ^c^	15 ± 1.0 ^b^	20 ± 2.0 ^a^	22 ± 2.0 ^a^
*L. monocytogenes*	−ve	−ve	11 ± 1.5 ^c^	22 ± 3.0 ^b^	23 ± 3.0 ^b^	26 ± 3.0 ^a^
Gram negative bacteria						
*E. coli*	−ve	11 ± 1.5 ^c^	14 ± 2.0 ^c^	20 ± 2.0 ^b^	24 ± 4.0 ^b^	30 ± 5.0 ^a^
*K. pneumonia*	−ve	−ve	15 ± 3.0 ^c^	20 ± 2.0 ^b^	25 ± 5.0 ^a^	26 ± 3.0 ^a^
*P. aeruginosa*	−ve	10 ± 1.0	14 ± 2.0	23 ± 3.0	29 ± 6.0	30 ± 4.0
*B. oleracea* pigment						
Gram positive bacteria						
*S. aureus*	−ve	9 ± 1.0 ^c^	12 ± 2.0 ^c^	16 ± 2.0 ^b^	21±2.0 ^a^	25±3.0 ^a^
*St. pyogenes*	−ve	−ve	−ve	9 ± 0.5 ^b^	10±1.0 ^a^	11±1.0 ^a^
*L. monocytogenes*	−ve	−ve	9 ± 1.0 ^c^	12 ± 1.5 ^c^	23±3.0 ^b^	26±4.0 ^a^
Gram negative bacteria						
*E. coli*	−ve	−ve	−ve	10 ± 1.0 ^b^	16 ± 2.0 ^a^	17 ± 2.0 ^a^
*K. pneumonia*	−ve	−ve	−ve	−ve	11 ± 1.0 ^b^	13 ± 1.5 ^a^
*P. aeruginosa*	−ve	−ve	−ve	11 ± 1.2 ^c^	15 ± 2.0 ^b^	19 ± 2.5 ^a^

Means in the same row having different letters are significantly different (*p* ≤ 0.05).

**Table 5 molecules-24-04280-t005:** Antifungal activities of some plant pigments (2000 µg/mL) against pathogenic fungi using well diffusion assays.

Microorganismis	Inhibition Zone (mm)		
	*H. sabdariffa*	*B. oleracea*	*B. vulgaris*
*A. ochraceus*	45 ± 5.0 ^a^	20 ± 2.0 ^b^	−ve
*F. oxysporum*	40 ± 4.0 ^b^	22 ± 3.0 ^a^	−ve
*P. expansum*	35 ± 3.0 ^c^	−ve	−ve
*P. citrinum*	36 ± 3.2 ^c^	−ve	−ve

Means in the same column having different letters are significantly different (*p* ≤ 0.05). −ve: No inhibition zone.

**Table 6 molecules-24-04280-t006:** MICs of *H. sabdariffa* and *B. oleracea* pigments against pathogenic fungi using well diffusion assay.

Microorganisms	Inhibition Zone (mm)					
	0	100	200	300	400	500
*H. sabdariffa* pigment						
*A. ochraceus*	−ve	23 ± 2.0 ^c^	30 ± 3.0 ^bc^	34 ± 3.0 ^b^	35 ± 3.0 ^b^	40 ± 5.0 ^a^
*F. oxysporum*	−ve	12 ± 1.0 ^d^	15 ± 1.5 ^c^	20 ± 2.0 ^b^	25 ± 2.0 ^a^	26 ± 2.0 ^a^
*P. expansum*	−ve	9 ± 0.5 ^c^	20 ± 2 ^b^	21 ± 2.0 ^b^	28 ± 2.5 ^a^	32 ± 3.0 ^a^
*P. citrinum*	−ve	13 ± 1.5 ^d^	25 ± 3.0 ^c^	32 ± 3.0 ^b^	33 ± 3.0 ^b^	38 ± 4.0 ^a^
*B. oleracea* pigment						
*A. ochraceus*	−ve	−ve	−ve	−ve	13 ± 2.0 ^b^	15 ± 1.0 ^a^
*F. oxysporum*	−ve	−ve	−ve	9 ± 1.0 ^c^	11 ± 1.0 ^b^	18 ± 2.0 ^a^
*P. expansum*	−ve	−ve	−ve	−ve	−ve	−ve
*P. citrinum*	−ve	−ve	−ve	−ve	−ve	−ve

−ve: No inhibition zone. Mean in the same row having different letters are significantly different (*p* ≤ 0.05).

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
