# Peer review of "Antimicrobial Activity and Chemical Constitution of the Crude, Phenolic-Rich Extracts of Hibiscus sabdariffa, Brassica oleracea and Beta vulgaris"

_molecules, 2019, doi:10.3390/molecules24234280_

Round 1
Reviewer 1 Report
Dear Authors,
The MS is interesting topic but structure of presentation must be improved. It can be accepted for publishing after major revision.
Please, see comments bellow.
The title of MS is not suit to the aim of MS. The title is Antimicrobial Activity and Physicochemical characteristics of Hibiscus sabdariffa, Crude Phenolic Rich Extract: an in vitro study. But in the aim of study is written: In the current study, crude phenolic rich extract (CPRE) was isolated from different sources such as Hibiscus sabdariffa (H. sabdariffa), Brassica oleracea (B. oleracea) and Beta vulgaris (B. vulgaris) and some parameters such as total anthocyanin, total phenolic content and total flavonoids were estimated.
In Introduction part is not well explained why where choose such plant objects.
Table 1. Antibacterial activity of some plant pigments (2000 μg/mL) against pathogenic bacteria using agar well diffusion assay. Is not understandably. Which kind of plant pigments? Its better to devitalize
In structure of MS I would write firstly about Chemical characterization of isolated CPRE and qualitative analysis by Gas chromatography–mass spectrometry (GC-MS) of HSP and after about antimicrobial effects.
Why Gas chromatography–mass spectrometry (GC-MS) analysis of HSP was done just for Hibiscus sabdariffa? Maybe better don’t present it as its not enough useful.
Author Response
First reviewer |
||||
Dear Authors, The MS is interesting topic but structure of presentation must be improved. It can be accepted for publishing after major revision. Please, see comments bellow. |
||||
Thanks for the thoughtful comments of the first reviewer. We have replied to all of them. |
||||
The title of MS is not suit to the aim of MS.. The title is Antimicrobial Activity and Physicochemical characteristics of Hibiscus sabdariffa, Crude Phenolic Rich Extract: an in vitro study. But in the aim of study is written: In the current study, crude phenolic rich extract (CPRE) was isolated from different sources such as Hibiscus sabdariffa (H. sabdariffa), Brassica oleracea (B. oleracea) and Beta vulgaris (B. vulgaris) and some parameters such as total anthocyanin, total phenolic content and total flavonoids were estimated. |
||||
Based on the sound reviewer comment, the title has been modified into {Antimicrobial Activity and Chemical Constitution of the Crude Phenolic Rich Extracts of Hibiscus sabdariffa, Brassica oleracea and Beta vulgaris} |
||||
In Introduction part is not well explained why where choose such plant objects. |
||||
The last paragraph in the introduction was modified to explain the reason of selecting the plant objects. This paragraph is now as follows: Based on the potential high content of anthocyanins and other bioactive compounds, three plants growing in Egypt were selected for this study; Hibiscus sabdariffa (H. sabdariffa), Brassica oleracea (B. oleracea) and Beta vulgaris (B.vulgaris), as the sources for isolating the crude phenolic rich extract (CPRE). These extracts (CPRE) were analysed for total phenolics content and total flavonoids and evaluated for their antibacterial and antifungal activity by different methods. |
||||
Table 1. Antibacterial activity of some plant pigments (2000 μg/mL) against pathogenic bacteria using agar well diffusion assay. Is not understandably. Which kind of plant pigments? Its better to devitalize |
||||
Done The title of the Table has been changed to be more expressive of the data to: Table 1. Antibacterial activity of crude phenolic rich extract (CPRE) (2000 µg/mL) from three plants against pathogenic bacteria using agar well diffusion assay. |
||||
In structure of MS I would write firstly about Chemical characterization of isolated CPRE and qualitative analysis by Gas chromatography–mass spectrometry (GC-MS) of HSP and after about antimicrobial effects. |
||||
The MS structure has been reorganized accordingly. |
||||
Why Gas chromatography–mass spectrometry (GC-MS) analysis of HSP was done just for Hibiscus sabdariffa? Maybe better don’t present it as its not enough useful. |
||||
Actually, the extract of Hibiscus sabdariffa was particularly distinct and showed the highest antimicrobial activity (either against bacteria or fungi) among the three studied plant sources. So it would be useful to know more information about this plant and will support futures studies on this plant source. |
||||
|
||||
|
Reviewer 2 Report
It was a pleasure to review the Manuscript "Antimicrobial Activity and Physicochemical characteristics of Hibiscus sabdariffa, Crude Phenolic 4 Rich Extract: an in vitro study". Abdel-Shafi and colleagues did an in-depth phytochemical analysis and evaluated the antibacterial and antifungal potency of the extracts of Hibiscus sabdariffa, Brassica oleracea and Beta vulgaris. The manuscript is linear, clear and well written (however, in some places there are typing errors that should be corrected). Today there is a huge demand for new molecules to counteract bacterial resistance; this manuscript goes in that direction. I believe it is fundamental to have innovative and in-depth studies on molecules of natural origin with potential therapeutic applications such as the Abdel-Shafi and colleagues manuscript.
However, I have some comments to address to the authors to improve and clarify some parts of their manuscript.
1) In the text, I found some typing and/or punctuation errors, a better linguistic and stylistic revision might be necessary;
2) In the Introduction chapter, authors should discuss better in vitro/in vivo test for their phytoextracts;
3) The authors should discuss the in vitro and in vivo toxicity of the molecules extracted;
4) The authors should indicate who performed the botanical classification of the plants used;
5) The bibliographic collection is accurate, but some more recent references should be added. In particular:
- line 49, Together with reference 19, Authors should insert “Kumar A et al. Cannabimimetic plants: are they new cannabinoidergic modulators? Planta. 2019”.
- line 62, Together with reference 24, Authors should insert “Mastinu et al., Zeolite Clinoptilolite: Therapeutic Virtues of an Ancient Mineral, Molecules 2019”
Author Response
Second Reviewer |
|||
Comments and Suggestions for Author It was a pleasure to review the Manuscript "Antimicrobial Activity and Physicochemical characteristics of Hibiscus sabdariffa, Crude Phenolic 4 Rich Extract: an in vitro study". Abdel-Shafi and colleagues did an in-depth phytochemical analysis and evaluated the antibacterial and antifungal potency of the extracts of Hibiscus sabdariffa, Brassica oleracea and Beta vulgaris. The manuscript is linear, clear and well written (however, in some places there are typing errors that should be corrected). Today there is a huge demand for new molecules to counteract bacterial resistance; this manuscript goes in that direction. I believe it is fundamental to have innovative and in-depth studies on molecules of natural origin with potential therapeutic applications such as the Abdel-Shafi and colleagues manuscript. |
|||
We appreciate very much the thoughtful effort of the second reviewer and value very much his encouraging comments. |
|||
However, I have some comments to address to the authors to improve and clarify some parts of their manuscript. |
|||
We have carefully responded to his comments. |
|||
1) In the text, I found some typing and/or punctuation errors, a better linguistic and stylistic revision might be necessary; |
|||
This linguistic revision has been done all through the manuscript by Prof. Dr. Mahmoud Sitohy |
|||
2) In the Introduction chapter, authors should discuss better in vitro/in vivo test for their phytoextracts; |
|||
This was revised and modified when required. |
|||
3) The authors should discuss the in vitro and in vivo toxicity of the molecules extracted; |
|||
The toxicity of in vitro and in vivo toxicity of the molecules has been incorporated in the discussion and marked in yellow. |
|||
4) The authors should indicate who performed the botanical classification of the plants used; |
|||
The used plants were botanically classified by Dr. Samer Teleb, Taxonomy and Flora, Botany and Microbiology Department, Faculty of Science, Zagazig University, Egypt. |
|||
5) The bibliographic collection is accurate, but some more recent references should be added. In particular: - line 49, Together with reference 19, Authors should insert “Kumar A et al. Cannabimimetic plants: are they new cannabinoidergic modulators? Planta. 2019”. - line 62, Together with reference 24, Authors should insert “Mastinu et al., Zeolite Clinoptilolite: Therapeutic Virtues of an Ancient Mineral, Molecules 2019”
|
|||
Done in the manuscript. |
|||
|
|||
Round 2
Reviewer 1 Report
Dear Authors,
after revision MS in present form can be accepted for publishing.
Reviewer 2 Report
Now, manuscript has improved considerably. I believe it is ready for publication.